# The Dissemination of *Metarhizium brunneum* Conidia by Females of the Red Palm Weevil, *Rhynchophorus ferrugineus,* Suggests a New Mechanism for Prevention Practices

**DOI:** 10.3390/jof9040458

**Published:** 2023-04-09

**Authors:** Sabina Matveev, Victoria Reingold, Eden Yossef, Noa Levy, Chandrasekhar Kottakota, Guy Mechrez, Alex Protasov, Eduard Belausov, Nitsan Birnbaum, Michael Davidovitz, Dana Ment

**Affiliations:** 1Department of Plant Pathology and Weed Research, Plant Protection Institute, Agricultural Research Organization (ARO), Volcani Center, HaMaccabim Road 68, Rishon LeZion 7528809, Israel; sabmat94@gmail.com (S.M.); vickire@gmail.com (V.R.); eden.yossef@mail.huji.ac.il (E.Y.); noalevy8923@gmail.com (N.L.); chandrabiotech@gmail.com (C.K.); nitsanb@kkl.org.il (N.B.); 2The Robert H. Smith Faculty of Agriculture, Food & Environment the Hebrew University of Jerusalem, Rehovot 7610001, Israel; 3Department of Food Science, Institute of Postharvest and Food Sciences, Agricultural Research Organization (ARO), Volcani Center, HaMaccabim Road 68, Rishon LeZion 7505101, Israel; guyme@agri.gov.il; 4Department of Entomology and Nematology, Plant Protection Institute, Agricultural Research Organization (ARO), Volcani Center, HaMaccabim Road 68, Rishon LeZion 7528809, Israel; protasov@volcani.agri.gov.il (A.P.); davidovitz@agri.gov.il (M.D.); 5Plant Science Institute, Agricultural Research Organization (ARO), Volcani Center, HaMaccabim Road 68, Rishon LeZion 7528809, Israel; eddy@agri.gov.il

**Keywords:** microbial pest control, entomopathogenic fungi, conidia transmission, dry conidia formulation, confocal laser scanning microscopy, scanning electron microscopy, egg hatching, larvae mortality

## Abstract

Direct contact between the conidia of entomopathogenic fungi (EPF) and their host is a prerequisite to successful infection; the host can, therefore, be infected by both direct treatment and by transmission of fungal inoculum from infested surfaces. This unique characteristic makes EPF especially relevant for the control of cryptic insects. In the case of the red palm weevil (RPW) *Rhynchophorus ferrugineus*, the eggs and larvae are almost inaccessible to direct-contact treatment. The objective of the present study was to investigate the mechanism of conidia transmission from a treated surface to host eggs and larvae. Foam pieces infested with *Metarhizium brunneum* conidial powder, conidial suspension, or distilled water were used as a laying surface for RPW females. The number of eggs laid was not affected by the EPF treatments and ranged from 2 to 14 eggs per female. However, hatching rate and larval survival were significantly reduced in the conidial powder treatment, resulted in 1.5% hatching and no live larvae. In the conidial suspension treatment, 21% of laid eggs hatched, compared to 72% in the control treatment. In both *M. brunneum* treatments, females’ proboscis, front legs and ovipositor were covered with conidia. The females transferred conidia in both treatments to the laying holes, reaching up to 15 mm in depth. This resulted in reduced egg-hatching rate and significant larval mortality due to fungal infection. The stronger effect on egg and larval survival using dry conidia seemed to result from better conidial adhesion to the female weevil in this formulation. In future studies, this dissemination mechanism will be examined as a prevention strategy in date plantations.

## 1. Introduction

The red palm weevil (RPW), *Rhynchophorus ferrugineus* Olivier (Coleoptera: Curculionidae), is one of the most devastating pests of various palm species [1]. The weevils develop within the tree trunk, destroying its vascular system and eventually causing the collapse and death of the tree. The weevil originated in South-East Asia and is now widely distributed in Asia, Oceania, Africa, Europe and the Middle East and Israel [2]. Females oviposit in the splitting bark, at the base of young leaves, or in wounds on the leaves and trunks. In these oviposition sites, eggs are laid close to the surface in a hole the female drills in the plant tissue. The hatched larvae tunnel into and feed on the surrounding tissue, thereby destroying it [3]. Control of the RPW is thus essential, but encounters difficulties in two aspects: first, resistance towards chemical pesticides, and second, their delivery towards the eggs and larvae. Attempts to develop an integrated pest management, based on combination with nematodes, are promising, as their motility enables them to reach the concealed stages of the RPW [4].

In entomopathogenic fungi (EPF), direct contact between conidia and host is a prerequisite to successful infection via adhesion of conidia to the cuticle and subsequently penetration of it [5,6]. The host can be infected both by direct treatment and by transmission of fungal inoculum from treated surface, e.g., soil, bark, plant, or from infected insects or cadavers to untreated insects or to subsequent developmental stages via the generation of new conidia [7,8,9].

In the case of *R. ferrugineus*, most of the life cycle passes within the tree trunk, making the pest inaccessible to direct-contact treatment. Due to that, one can hypothesize that adults are the only stage that can be efficiently exposed to conidia and be infected upon emergence from the plant or upon contact with other individuals. Previously, we have found that up to 42% of adult weevils trapped from various locations in Israel were infected with Hypocrealean EPF, *Beauveria* or *Metarhizium* spp. [10]. That observation supports the existence of some rout of infection. In another previous study, the authors demonstrated that an Israeli *M. brunneum* (previously designated as *M. anisopliae* 7) isolate caused 80% mortality in larvae of the RPW. Further, they speculated that females can vector conidia to their progeny via contaminating the eggs during oviposition [11]. However, results could not support that speculation due to low mortality rates of eggs, probably due to insufficient inoculum transfer.

In the present study, we hypothesized that the application of EPF conidia to surfaces on which RPW females oviposit would result in the acquisition of conidia to the body surface of the females, and transmission and dissemination of conidia into the region in which eggs are being laid. Consequently, this transmission would result in direct contact between conidia and eggs. To evaluate this, we developed an experimental setup system based on the highly efficient *M. brunneum* isolate applied as dry or wet conidia.

This system enabled us to evaluate whether the female can effectively acquire conidia, either in dry or wet formulation, and convey it onwards to its progeny and, if so, how the dissemination occurs. The specific objectives of this study were: (1) to develop a procedure to examine the transfer of wet and dry conidia from a treated surface to *R. ferrugineus* eggs and larvae; (2) To monitor the distribution of conidia on a treated surface following oviposition behavior of *R. ferrugineus* females; (3) To examine the effect of conidia-treated surface on *R. ferrugineus* females’ survival, eggs per female, egg hatch rate and larvae survival.

## 2. Materials and Methods

### 2.1. Entomopathogenic Fungi

*Metarhizium brunneum*-7 constitutively expressing GFP reporter gene (Mb7-GFP) [12,13] was grown on Sabouraud dextrose agar (SDA; Difco) for two weeks at 28 ± 0.5 °C. For conidia stock preparation, Mb7-GFP was grown on rice as described previously [14,15]. Conidia were separated from the rice by sieving through a 12-mesh sieve (1.68 mm). Harvested conidia were maintained at 4 ± 0.5 °C in a sealed plastic box. For conidia suspension preparation, conidia were put into polypropylene tubes with sterile distilled water containing 0.01% Triton X-100. The suspensions were vortexed and filtered through Miracloth (Calbiochem; La Jolla, CA, USA), and spore concentrations were determined with a hemocytometer. Suspensions were adjusted to the required conidial concentrations as described in Section 2.3 in 0.01% Triton X-100, and the percentage of viable conidia was determined on SDA prior to each bioassay (germination assay). Only conidial suspensions with at least 95% germination were taken for bioassays.

### 2.2. Insects

Red palm weevil adults were trapped by Picusan^®^ traps (SANSAN PRODESING SL, Valencia, Spain) contain pheromone-kairomone lures (4-methyl-5-nonanol and 4-methyl-5-nonanone, ethyl acetate and sugar molasses, supplied by Biobee, Sde Eliyahu, Israel) set in different locations along the central coastal plain of Israel. The trapped weevils were mated in 3:2 female-male ratio for one week before the experiments and were kept in groups of up to 50 adults in plastic boxes (20 cm wide × 40 cm long × 20 cm high). Each box had four 10 cm diameter mesh-covered holes for ventilation. The insects fed on sugarcane pieces. Boxes were kept in environment control chambers at 25 ± 1 °C and 70 ± 10% relative humidity and 10:14 hr light/dark photoperiod.

### 2.3. Laboratory Bioassay

Mated females were transferred individually to 500 mL plastic boxes with ventilated caps for five days. A piece of floral pre-wetted foam (green Styrofoam; Arizot Plus, Hulon, Israel), herein foam pieces (5 cm wide × 5 cm long × 1 cm high) and a piece of sugarcane (1 cm^3^, to serve as a food source) were placed into each box (Figure 1A). For the control treatment, the foam pieces were soaked in distilled water for 5 s. For the conidia suspension treatments, the foam pieces were soaked in 5–7 mL of 10^8^ conidia/mL suspension. For the conidia powder treatments, the pre-wetted foam pieces were rubbed on all sides with 0.5 g of conidia (equivalent to total of 5 × 10^8^ conidia). Boxes were kept in the chambers at similar conditions as in Section 2.2 for up to two weeks. Five days after the onset of experiment, the females were removed from the boxes and eggs were counted by examining the foam pieces for the typical oviposition holes made by the females. To count eggs and for further observations, the foam pieces were cut gently by sterile scalpel knife and eggs were removed by sterile needle and counted for each replicate. After counting, eggs were returned back to their respective foam pieces to monitor hatching. During the experiment period, egg hatching was monitored for up to 13 days post inoculation (DPI) and the larvae that hatched in the foam pieces were counted. Dead larvae or unhatched eggs were surface sterilized and incubated in darkness at 25 ± 0.5 °C in a 55-mm Petri dish lined with moist filter paper and sealed with Parafilm to evaluate mycosis. Each treatment consisted of 10 females. Each experiment was repeated at least twice.

### 2.4. Sampling of Foam Pieces, Eggs and Females for Microscopic Observations

This section describes sample preparation for confocal laser scanning microscopy and scanning electron microscopy examination of foam pieces, eggs and females. The same experimental system described above was used (Figure 1A, Section 2.3). Treatments included distilled water as negative control, conidia suspension, and conidia powder all applied as described in Section 2.3. In addition, for each of the treatments, 10 microcosms were incubated without female weevils inside to create undisturbed microcosms. Each treatment consisted of 10 females. The microcosms were incubated for five days. Each experiment was repeated twice.

Sampling of foam pieces—Foam pieces were sampled from microcosms with females and from undisturbed ones. From the surface of the foam pieces of each treatment, the upper part was cut at about 1.5 cm thick and 1 cm^2^ surface area. Additionally, from microcosms with females, the hole created by the oviposition was cut out entirely with the surrounding foam. These holes were arbitrarily divided into three sections, herein A, B, C, as described in Figure 1B. Each section was about 5 mm thick. The various sections were immediately prepared for microscopy observations as described below.

Sampling eggs and females—During the sampling of the foam pieces and hole collection, eggs from the different treatments were collected and transferred to a 55 mm Petri dish lined with moist filter paper for microscopy observation. Females from each treatment were transferred to clean 55 mm Petri dishes and anaesthetized by 30 min incubation in −20 °C. The females were dissected by sterile scalpel. The front legs, mouth parts (proboscis), antennae and ovipositor from each female were collected and transferred to a 55 mm Petri dish lined with moist filter paper for microscopy observation (Figure 1B).

### 2.5. Confocal Laser Scanning Microscopy and Image Analysis

Live imaging of the freshly collected samples was carried out by confocal laser scanning microscopy (CLSM) to evaluate the distribution of the conidia on the foam pieces and the dissemination of conidia from the foam pieces to females’ organs and eggs. Soon after its collection, samples of foam, eggs and females were placed on glass slides. The samples were observed by CLSM (Olympus, Fluoview 500) using Argon laser 488 nm excitation. Fluorescence emission of GFP was recorded at 500–520 nm. For 3D images, acquisition used a Leica SP8 laser scanning microscope (Leica, Wetzlar, Germany) equipped with a solid-state laser with 488 nm light, HC PL APO CS 20x/0.75 objective (Leica, Wetzlar, Germany) and Leica Application Suite X software (LASX, Leica, Wetzlar, Germany).

For Mb7-GFP fluorescent intensity measure, image stacks were first projected using a Z projection (at maximum intensity) to find all the fluorescent conidia, then intensity level was set to 23 as default and measured using ImageJ software [16]. Image analysis was done for the foam piece samples in order to evaluate the GFP intensity level as a quantitative measurement of conidia in the various treatments. To determine the local maxima in every image, the noise tolerance level was set to 23 for all images and ‘count’ was selected as output in ImageJ software.

### 2.6. Scanning Electron Microscopy

Scanning Electron Microscopy (SEM) observations were performed using a MIRA3 field-emission SEM microscope (Tescan, Brno/Czech Republic) with an acceleration voltage of 1.0 kV and a secondary electron (SE) detector. Prior to imaging, a thin layer of carbon was evaporated onto the samples to render them electrically conductive, and to avoid surface charging by the electron beam.

### 2.7. Data Analysis

All statistical analyses were performed using JMP^®^ Version 14 (1989–2019) software (SAS Institute Inc., Cary, NC, USA). Results are presented as mean ± SE of replicate analyses and are either representative of or include at least two independent experiments. Means of replicates were subjected to statistical analysis and considered significant when *p* ≤ 0.05.

The total number of eggs was counted on the fifth day for each treatment and subjected to analysis of variance (ANOVA). Egg-hatching rates were monitored from day 5 and over a period of two weeks in the different treatments. The square roots of the hatched eggs proportion were arcsine-transformed and then subjected to repeated-measures ANOVA to examine the effects of the different treatments, time of observation and the Treatment × Time interaction. The initial number of eggs was used as a weighting variable.

The intensity level as a measure for amount of GFP expressing conidia in the different treatments was subjected to ANOVA by the standard least square model, followed by multiple comparison for full factorial. If effect was found to be significant (*p* < 0.05), Student’s-t all pairwise comparison was used for comparisons among means for the conidia ‘count’ data obtained by the image analysis of the CLSM micrographs.

## 3. Results

### 3.1. Microscopy Assessment of Mb7-GFP Conidia Acquisition by Females and Dissemination from Surface of Foam Pieces to Laying Holes

To assess conidia acquisition by females and dissemination from surface of foam pieces to the laying holes, undisturbed foam pieces treated with either conidia suspension or conidia powder were sampled 5 DPI. In parallel, foam pieces with either of the treatments were sampled five days after female weevils were released into the microcosms and laying holes were confirmed. The foam pieces sampled were observed under CLSM and SEM at three sections, as described in Figure 1B. In undisturbed foam pieces, Mb7-GFP conidia were observed on section A, the outer surface, of either of the treatments (Figure 2A,C), but not in section B or C (Figure 2E,G,I,K). In the laying holes, conidia were observed in all three sections in either of the treatments by both CLSM and SEM (Figure 2B,D,F,G,J,L and Figure 3).

To evaluate the mechanism by which conidia were transmitted into the laying holes during female weevil egg laying, eggs, proboscis (Figure 4A,B) and front legs (Figure 4C,D) were observed by CLSM and proboscis (Figure 5A,B), front legs (Figure 5C,D) and ovipositor (Figure 5E,F) by SEM. Conidia were observed on all inspected females’ organs for both fungal treatments of conidia powder and conidia suspension.

Dissemination of conidia from the treated surface into the laying hole was also validated by the intensity levels of Mb7-GFP fluorescence before the entry of the weevil into the microcosms and after entry and laying activity. The intensity levels were significant (ANOVA: F = 4.28, DF = 11, *p* < 0.005) and affected by the laying activity of the weevil and the inspected section in the laying hole (Effect tests: section F = 11.7, DF = 2. *p* < 0.0005; weevil × section interaction F = 10.13, DF = 2, *p* < 0.001). Before entry of the weevil into the microcosms, the highest intensity was measured in the outermost section A compared with the inner sections B and C (Figure 6). The type of formulation did not affect the intensity levels (*p* = 0.169). Upon entry by the weevil, significant change was observed in the intensity level in all sections compared with section A prior to weevil entry. The intensity levels increased significantly in similar rates in sections B and C at the laying hole in either of the formulations (all pairwise comparison for weevil × section A × treatment by Student’s t, powder *p* = 0.0249, suspension *p* < 0.001; Figure 6).

### 3.2. Susceptibility of R. ferrugineus Females, Eggs and Larvae to Mb7-GFP under Laboratory Conditions

As it was demonstrated that the females disseminate conidia to the laying hole, we further evaluated the effect of this interaction on females’ survival, fecundity as number of eggs laid, egg hatching rate and the presence of conidia on eggs and larvae. *R. ferrugineus* female mortality was not detected throughout the experiments in any of the treatments. The number of eggs laid in the oviposition boxes over a period of five days ranged from 5 to 12 per female and with no significant difference between treatments (ANOVA: F = 0.7, DF = 2, *p* = 0.52; Figure 7A). In general, egg hatching was observed from day 6 of the experiment onset through day 13. The maximal egg hatching rates were 72% in the control group, 21% in the conidial-suspension treatment and 1.5% for the conidial-powder treatment (Figure 7B). The hatching rate varied significantly according to the treatment, as the conidia-powder treatment significantly reduced hatching rate compared to control and conidia suspension (Univariate analysis of hatching rate by DPI, F = 17.94, DF = 2, *p* = 0.0022; Figure 7B). The DPI effect and its interaction with the Treatment effect were significant as well (effect test by whole-model analysis: Time—F = 9.78, DF = 4, *p* < 0.0001; Treatment × DPI—F = 4.74, DF = 8, *p* < 0.001).

Eggs from the control group did not exhibit signs of mycosis (Figure 8A) and Mb7-GFP conidia were not detected under CLSM (Figure 8E). All non-hatched eggs in both Mb7-GFP treatment exhibited signs of mycosis and sporulation (Figure 8B,C). Mb7-GFP conidia were detected on all eggs from both conidia suspension (Figure 8F) and conidia powder (Figure 8G) treatments, with a higher amount of observed conidia in the powder treatment. All larvae hatched in the control group survived, but larvae hatched in both Mb7-GFP treatments exhibited signs of mycosis and sporulation (Figure 8D), confirmed under CLSM as Mb7-GFP (Figure 8H).

## 4. Discussion

Capturing the essential behavioral processes that occur between arthropod host and pathogen is essential to exploiting entomopathogens for microbial pest control [17]. As a general rule, transmission is described as the number of new infections per unit of time, a factor that depends on parasite transmission efficiency and the density of both parasites and hosts [18]. The dispersal of pathogen propagules—EPF conidia—from infected to new hosts is a prerequisite for disease initiation and pathogenic interactions [19]. Horizontal transmission usually refers to the transmission of pathogen propagules ‘within generations’ (also termed autodissemination) or between individuals of the same host species, whereas vertical transmission occurs ‘between generations’ [17]. The transmission mechanism of EPF conidia from the female to its progeny has never been studied; we, therefore, aimed to elucidate its mechanism and impact in RPW. We describe the acquisition and dissemination of *M. brunneum* conidia (‘the parasite’) from a conidium-treated surface to the female RPW (‘the transmitter’) and from there, via the female’s laying activity, to the egg laying hole and to the eggs and larvae (‘the hosts’).

Dissemination and transmission of EPF conidia is affected by numerous insect behaviors, among them movement, grooming, foraging, avoidance, mating and oviposition. Possible transmission due to oviposition behavior has been documented in only one study on long-horned beetles [20], where transmission of *Metarhizium anisopliae* propagules during oviposition of *Anoplophora glabripennis* was discussed. In nature, arthropods may accumulate or acquire the conidia themselves from naturally occurring EPF, whereas in an agricultural scenario, this acquisition may be due to exposure to EPF-covered surfaces (e.g., soil, foliage, bark) [21,22]. Either way, the consequence may be transmission of conidia between males and females (see, for example [9,23,24,25]). Studies on horizontal transmission in weevils (Coleoptera: Curculionidae) are scarce. Gottwald and Tedders [26] measured the distance of colonies of *B. bassiana* and *M. anisopliae* growing in a soil environment from infected *Curculio caryae* (pecan weevil) cadavers and its effect on conidial transmission. A more recent study [23] evaluated horizontal transmission of *M. anisopliae* and *B. bassiana* to adults of *Kuschelorhynchus macadamiae* from other infected adults and conidiated cadavers. Interestingly, high rates of conidial transmission were found, but mortality rates depended on the environmental conditions [23]. Variation among *M. anisopliae* and *B. bassiana* isolates was examined with respect to mortality rates due to horizontal transmission between adults of the banana root borer *Cosmopolites sordidus* [24].

Previous studies on the interactions between *R. ferrugineus* and EPF have mainly evaluated virulence and efficacy toward adults and larvae by screening various EPF species and strains (reviewed in [27]). With respect to horizontal transmission, it was shown that adult weevils in contact with EPF conidia disseminate those conidia among the adults [27,28,29,30]. Though the dissemination of conidia between females and their progeny was not proven, Gindin et al. [11] proposed that RPW females that are contaminated with conidia may infect their progeny via egg contamination during the egg-laying process, but they were not able to prove it. They suggested that the futility of these infection attempts might be due to low inoculum levels, the oviposition medium used (sugarcane) or the use of a fungal formulation that was not appropriate for conidial transmission (i.e., spray application or contact with spores on the culture media). In the current study, we established an experimental setup that enables precise evaluation of conidia acquisition and dissemination and contamination of laying hole by: (a) applying higher inoculum levels (5 × 10^8^ conidia mL^−1^, compared to 2–5 × 10^7^ conidia mL^−1^), which maximized the chances of a large number of conidia being acquired and disseminated by the females; (b) using an inert material (foam) for the oviposition site, which enabled straightforward observation of the laying holes, eggs, larvae and conidia, and eliminated any possible inhibitory effects of nutrients in the egg-laying medium on host–pathogen interactions. These modifications, followed by microscopy observations, enabled verifying conidial acquisition by the females and disseminating it from the surface into the laying holes and to the eggs.

The dissemination of conidia from the oviposition site to the egg-laying hole and then to the eggs, via the female’s laying activity, was evident and pronounced. This phenomenon resulted in a significant reduction in egg-hatching rates and increased larval mortality due to mycosis, while the adult females remained alive. In accordance with our results, previous studies found no effect of EPF treatment on the number of eggs laid by females [11,31,32]; although conidia adhered to the female weevil body parts, its fecundity was unaffected and mycosis was not observed. We suggest that either there was not enough inoculum to initiate such sublethal effects, or the abiotic conditions were not favorable for mycosis in the females.

The type of fungal conidial preparation had a significant effect on hatching rates. Treating the oviposition surface with dry conidia resulted in lower egg hatch than treatment with a conidial suspension, but the microscopic evidence could not confirm higher dissemination rates. Similar observations have been made previously, where a dry conidial formulation resulted in higher transmission rates than inoculation with wet conidia for *Ceratitis capitata* adults [25]. Moreover, high efficacy of a sporulated-rice formulation was observed in field trials involving RPW [33] and the black palm weevil *Rhynchophorus bilineatus* [34]. Notably, in our study, dry and wet conidia originated from complete media; however, some differences may be attributed to the variation in growth conditions. In *Aspergillus fumigatus*, genes encoding adhesion factors were highly expressed in dormant dry conidia; these adhesion factors play a role in conidial hydrophobicity and high adherence rate [35]. Thus, better acquisition and adhesion of fungal conidia in a dry formulation are likely the result of the conidia’s surface biochemistry characteristics, which may be regulated to some extent by adhesion related genes.

The results stemming from the current study not only prove conidial acquisition and dissemination by female weevils and their effect on egg-hatching rates and larval survival in RPW, but also determine the underlying mechanism of transmission from treated surface to the female weevils and their progeny. Conidial adhesion is a non-specific process, and it is not surprising that females were entirely covered by conidia [36]. During oviposition, the female weevil moves around on the oviposition surface in search of a spot to lay the egg. During that search, the female’s legs are in contact with the conidia, which readily adhere to them. Once a suitable laying site is located, the female drills a hole using its mouthpart, resulting in further acquisition of conidia. As the mouthparts are pushed into the laying holes, the conidia are transmitted into it, as evidenced by the presence of conidia at up to a 15 mm depth following drilling of the holes and by the change in fluorescent intensity before and after drilling. The ovipositor is covered with conidia as well, thereby facilitating further conidial dissemination into the laying hole.

## 5. Conclusions

We examined and evaluated the transmission mechanism of *M. brunneum* conidia from a treated surface to female RPW, and from there to the laying hole, the eggs and larvae. Consequently, the acquisition of the conidia by the RPW female led to mycosis in eggs and larvae. We demonstrate dissemination of conidia from the surface into the laying hole by the female mouthpart and ovipositor during egg-laying activity, resulting in pronounced egg and larval mortality. A dry formulation of *M. brunneum* conidia was more efficient at reducing egg hatch and larval survival, suggest better adhesion to the females. These results encourage the further evaluation of this unique mechanism of EPF dissemination into the egg-laying hole as a preventative measure in RPW management. We consider this mechanism as a venue to tackle other woodborers such as long-horned beetles, of which management is a challenge.

## Figures and Tables

**Figure 1 jof-09-00458-f001:**
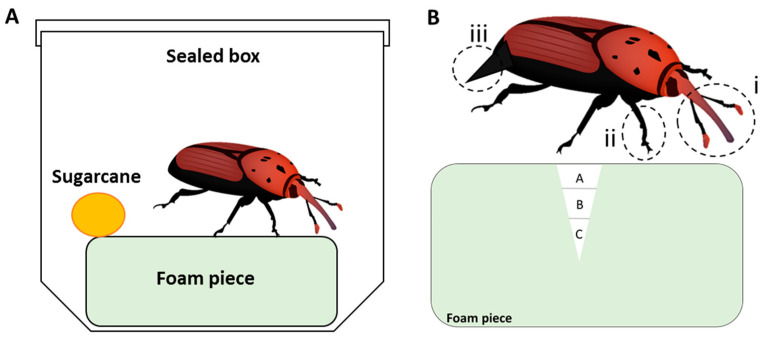
Schematic representation of the experimental microcosm used to evaluate the Mb7-GFP conidia acquisition and dissemination from foam piece to *Rhynchophorus ferrugineus* female and eggs, and the susceptibility of eggs and larvae to Mb7-GFP. (**A**) Each microcosm included a single mated female, a foam piece and a piece of sugarcane. (**B**) Schematic representation of a foam piece with a hole made by the female. Laying hole divided into three sections of 5 mm each: A, B, C, as depicted. (i) Mouth part (proboscis) and antennae, (ii) front leg, (iii) Ovipositor.

**Figure 2 jof-09-00458-f002:**
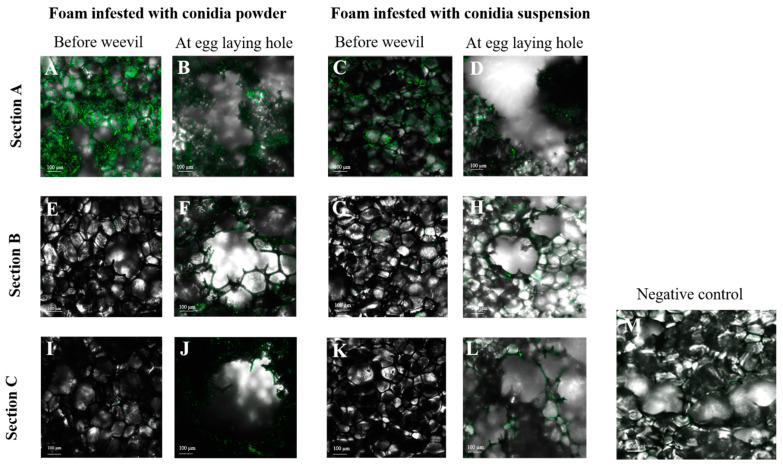
Confocal laser scanning micrographs of Mb7-GFP conidia-treated foam pieces before and after weevil entry in three depth sections (sections A–C indicated at the left side). Foam treated with conidia powder 5 × 10^8^ total conidia—before weevil entry (**A**,**E**,**I**) and After weevil entry (**B**,**F**,**J**). Foam treated with conidia suspension 10^8^ conidia mL^−1^—before weevil entry (**C**,**G**,**K**) and after weevil entry (**D**,**H**,**L**). Non-treated foam, negative control (**M**) Scale bar = 100 µm.

**Figure 3 jof-09-00458-f003:**
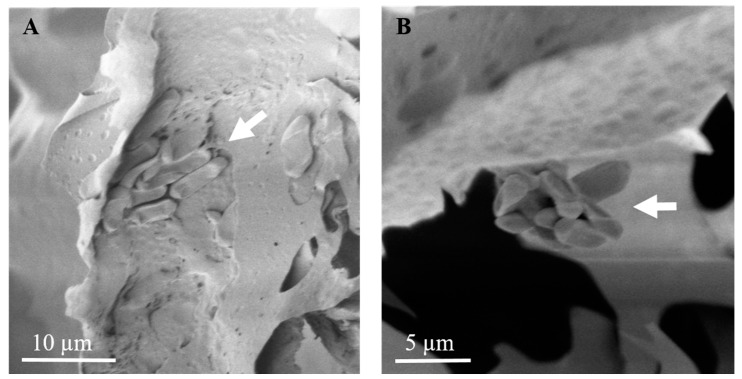
Scanning electron micrograph of treated foam pieces with (**A**) Mb7-GFP conidia powder 5 × 10^8^ total conidia and (**B**) conidia suspension 10^8^ conidia mL^−1^, within laying holes at section C. Scale bar designated in each micrograph. Arrows indicate conidia clumps in each treatment.

**Figure 4 jof-09-00458-f004:**
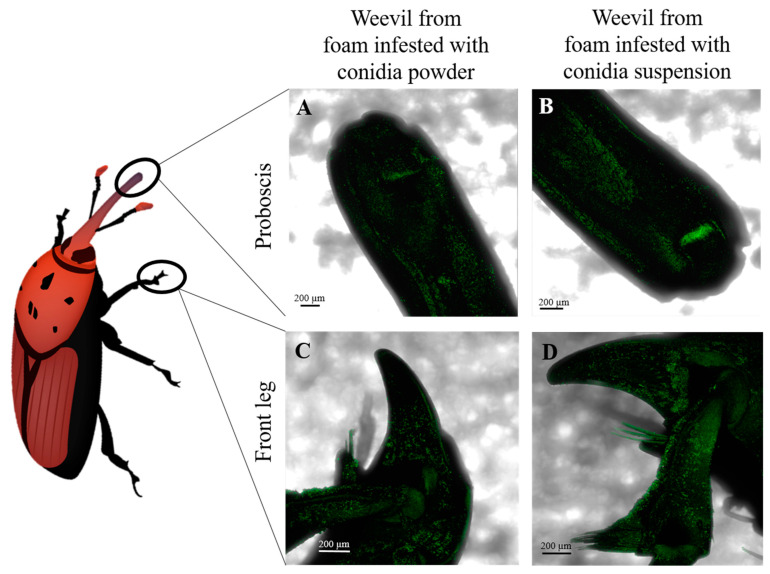
Confocal laser scanning micrographs of female weevil proboscis (**A**,**B**) and front leg (**C**,**D**) after exposure to foam treated with Mb7-GFP conidia powder, 5 × 10^8^ conidia in total (**A**,**C**) or Mb7-GFP conidia suspension 10^8^ conidia mL^−1^ (**B**,**D**). Scale bar = 200 µm.

**Figure 5 jof-09-00458-f005:**
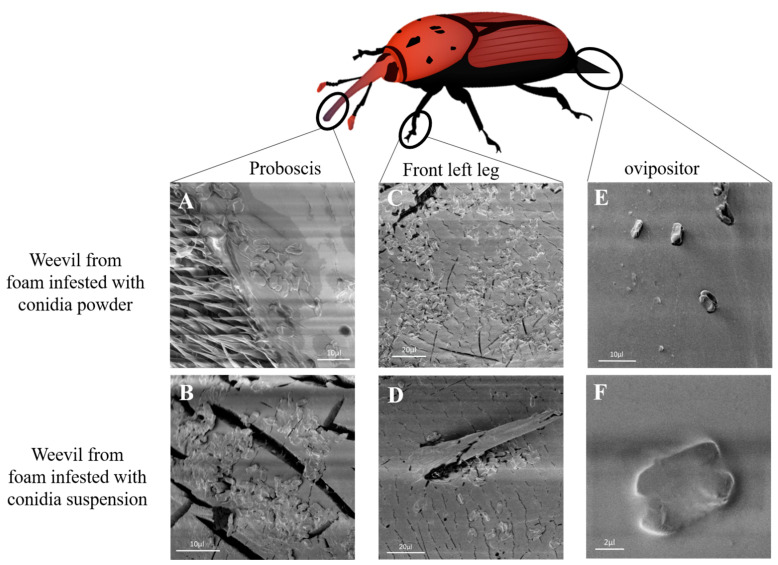
Scanning electron micrograph of female weevil proboscis (**A**,**B**), front leg (**C**,**D**) and ovipositor (**E**,**F**) after exposure to foam treated with Mb7-GFP conidia powder 5 × 10^8^ conidia total (**A**,**C**,**E**) or foam treated with conidia suspension 10^8^ conidia mL^−1^ (**B**,**D**,**F**). Scale bar designated for each micrograph.

**Figure 6 jof-09-00458-f006:**
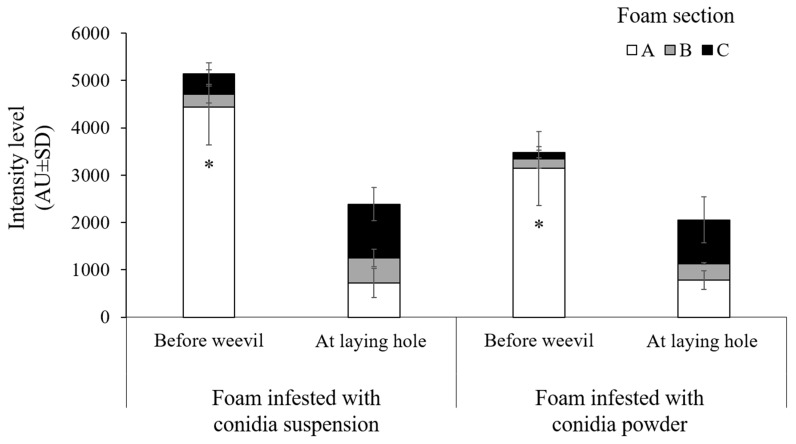
Dissemination of conidia from surface into laying hole. Intensity level (arbitrary units AU ± SD) measured by ImageJ software of Confocal laser scanning micrographs of Mb7-GFP-treated foam pieces with conidia suspension (10^8^ conidia mL^−1^) or conidia powder (5 × 10^8^ conidia total) before weevil entry to the microcosms and after five days from weevil entry, at laying holes. Three to four individual sections of foam pieces were analyzed**.** Asterisk indicates significant difference of intensity level measured in section A before weevil entry compared with all other sections and at weevil laying holes, * *p* < 0.0005 by all pairwise comparison and post hoc analysis by Student’s t.

**Figure 7 jof-09-00458-f007:**
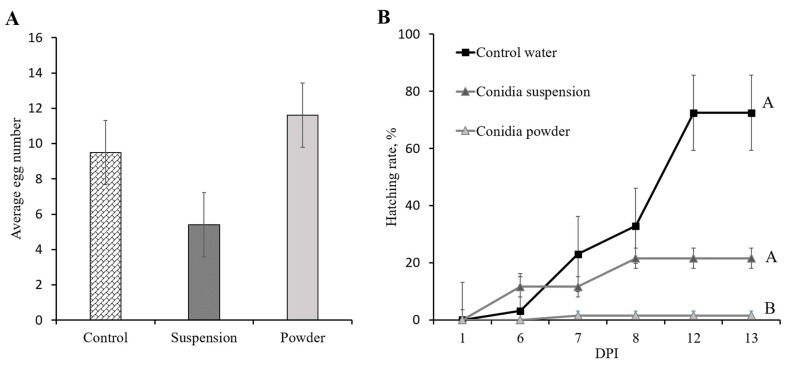
Assessment of Mb7-GFP’s effect on *Rhynchophorus ferrugineus* eggs per female and egg hatching rate by microcosms assay. (**A**) Average number of eggs per female in non-treated control and following Mb7-GFP conidia treatment as conidia suspension (10^8^ conidia mL^−1^) and conidia powder (5 × 10^8^ conidia total). (**B**) Average hatching rate of eggs in the control group and following Mb7-GFP conidia treatment as conidia suspension and conidia powder. Curve followed by different letter differ significantly. Univariate analysis of hatching rate by days PI (Repeated measures)—Treatment *p* = 0.0022; Day PI *p* < 0.0001; Treatment × Day *p* < 0.001. PI = Post-Inoculation.

**Figure 8 jof-09-00458-f008:**
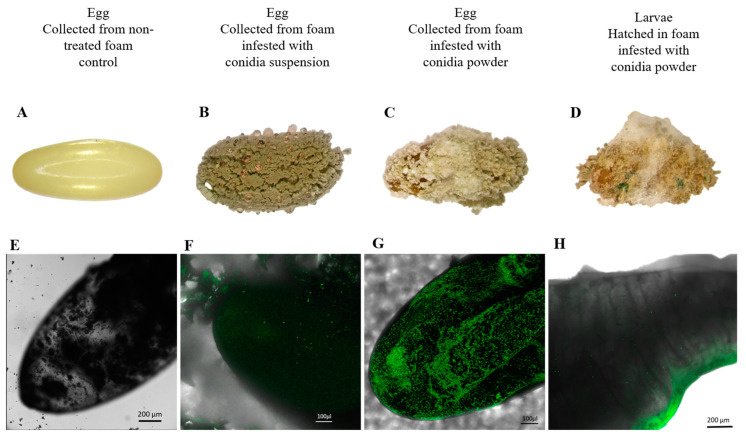
*Rhynchophorus ferrugineus* eggs and larvae sampled from foam pieces at 5 DPI. Observation for mycosis and sporulation in control egg (**A**,**E**), egg collected from foam treated with Mb7-GFP conidia suspension (10^8^ conidia mL^−1^) (**B**,**F**); egg collected from foam treated with Mb7-GFP conidia powder (5 × 10^8^ conidia total) (**C**,**G**) and larvae collected from foam treated with Mb7-GFP conidia powder (5 × 10^8^ conidia total) (**D**,**H**). (**A**–**D**) Micrographs taken under binocular microscopy following incubation for sporulation. (**E**–**H**) Micrographs taken under confocal laser scanning microscopy of eggs and larvae collected from foam pieces. Scale bar = 200 µm.

## Data Availability

Correspondence and requests for materials should be addressed to D.M.

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
