# Peer review of "The Dissemination of Metarhizium brunneum Conidia by Females of the Red Palm Weevil, Rhynchophorus ferrugineus, Suggests a New Mechanism for Prevention Practices"

_jof, 2023, doi:10.3390/jof9040458_

Round 1
Reviewer 1 Report
The contribution demonstrated the dissemination of the entomopathogenic fungus Metarhizium brunneum by females of Rhynchophorus ferrugineus under controlled conditions.
The authors designed a test to demonstrate their hypothesis. The paper is generally interesting and well-prepared.
Some aspects to consider in the contribution are:
1. Why did the authors use the fungus Metarhizium brunneum? Justify it in the introduction.
2. Why did you use conidia suspension and powdered conidia? Not indicated in the objectives
3. Have you cultured the conidia that infected the eggs to verify that they were inoculated?
4. Scientific names are in italics, check and correct them in the manuscript's text and the references section.
5. Figure 1: Indicates the measurement of the foam piece and sections A, B and C.
6. Figure 2. This could be improved; the fungal structures are not visible.
7. Figure 3. points out the conidia in the images.
8. Adjust the references and the manuscript according to the Journal of Fungi.
Author Response
Reviewer 1
Comments and Suggestions for Authors
The contribution demonstrated the dissemination of the entomopathogenic fungus Metarhizium brunneum by females of Rhynchophorus ferrugineus under controlled conditions.
The authors designed a test to demonstrate their hypothesis. The paper is generally interesting and well-prepared.
Some aspects to consider in the contribution are:
- Why did the authors use the fungus Metarhizium brunneum? Justify it in the introduction.
We have used the isolate of M. brunneum, as it found efficient towards the weevil in a previous study by Gindin et. al 2006. We added the explanation in lines 63 and 73.
- Why did you use conidia suspension and powdered conidia? Not indicated in the objectives
Using wet and dry conidia provided a larger comparison, and increased our knowledge on the efficient dissemination of conidia. We demonstrated the better activity of dry conidia and proposed an explanation to this observation. We have added this point to our objectives (in lines 74 and 76).
- Have you cultured the conidia that infected the eggs to verify that they were inoculated?
All experiments included a sporulation assay of the unhatched eggs. All unhatched eggs demonstrated Metarhizium sporulation after incubation, and thus provide the evidence of successful inoculation (indicated in lines 272-274).
- Scientific names are in italics, check and correct them in the manuscript's text and the references section.
Thank you for pointing, we have corrected the text accordingly. (e.g. line 61, 428, 432 etc.)
- Figure 1: Indicates the measurement of the foam piece and sections A, B and C.
We added to the legend of the figure the size of the sections (5mm; line 132)
- Figure 2. This could be improved; the fungal structures are not visible.
Yes, the structures are not visible in these figures, due to the magnification used. Unfortunately, we do not have higher magnification figures from these experiments, and enlargement resulted in low quality. To address the issue, we have added a negative control figure of the non-treated foam, and increased the intensity of all figures (both treatments and control) using the same values (of brightness +40 and contrast +20). The GFP is now better visualized and the green artifacts may be observed in the negative control. It is clear that the green intensity of control sample (using the same criteria) is much less, compared to the treatments (real GPF), indicating the presence of fungal-GFP in the treatments and not in the control.
- Figure 3. points out the conidia in the images.
We added an arrow in each image indicating the presence of conidial clumps, and designated it in the figure legend (line 219).
- Adjust the references and the manuscript according to the Journal of Fungi.
Thank you, the references were corrected accordingly. Please note that it was NOT done with track-change to avoid over correction on the MS.

Reviewer 2 Report
This manuscript described a dissemination of fungal conidia from treated surface to eggs and larvae. Results obtained from these experiments are interested. However insect control using dried conidia spore is not new. The manuscript should be revised for clarity and following points should be addressed.
1. Title is quite long and the words "suggesting a new mechanism for prevention practices" is misleading. Applying dried spore and spore suspension to cover surface where insect resides e.g. plant parts and soil is a normal practice. This method has been used for many entomopathogenic fungi such as Beauveria bassiana, Metarhizium anisopliae, Purpureocillium lilacinum, Cordyceps fumosorosea, etc.
2. In abstract (lines 31-32) and conclusions (lines 393-394), authors concluded that dry spore was more effective than spore suspension. It might not be true for this experiment since spores from both formulations were prepared from different media, SDA for suspension and rice for dry spore. It is possible that culture media compositions contribute to virulence. It should be better to prepare spore that grow from the same medium e.g. on rice.
3. Labelling on some figure e.g. Fig 4, 5 & 8 is confusing. What is Sponge powder? Sponge suspension? Suspension egg Powder egg? Powder larvae?
Author Response
Reviewer 2:
Comments and Suggestions for Authors
This manuscript described a dissemination of fungal conidia from treated surface to eggs and larvae. Results obtained from these experiments are interested. However insect control using dried conidia spore is not new. The manuscript should be revised for clarity and following points should be addressed.
- Title is quite long and the words "suggesting a new mechanism for prevention practices" is misleading. Applying dried spore and spore suspension to cover surface where insect resides e.g. plant parts and soil is a normal practice. This method has been used for many entomopathogenic fungi such as Beauveria bassiana, Metarhizium anisopliae, Purpureocillium lilacinum, Cordyceps fumosorosea, etc.
We have modified the title of the manuscript
- In abstract (lines 31-32) and conclusions (lines 393-394), authors concluded that dry spore was more effective than spore suspension. It might not be true for this experiment since spores from both formulations were prepared from different media, SDA for suspension and rice for dry spore. It is possible that culture media compositions contribute to virulence. It should be better to prepare spore that grow from the same medium e.g. on rice.
This is indeed a good point and this potential explanation added to the discussion (lines 368-369). It is important to state, that the rice itself has been inoculated with the conidia collected from SDA plates, and both represent a saprophytic development of Metarhizium in a rich media, and this may reduce potential variation in pathogenicity related genes.
- Labelling on some figure e.g. Fig 4, 5 & 8 is confusing. What is Sponge powder? Sponge suspension? Suspension egg Powder egg? Powder larvae?
Thank you for this comment. We have changed the legends to fit better the main text, material and methods and results. Within the figures, text corrected accordingly.

Reviewer 3 Report
The manuscript presents very nice study on interaction between one of the most serious pest of date and entomopathogenic fungi. The idea to apply conidia of fungus in the way that beetle female can transport them into place where it lays eggs is clever. Results are convincing and open new strategy for biocontrol of this pest. The manuscript for sure fits well into JoF scope, is well written, thought some parts can be improved. See comments below.
Specific comments (numbers indicate lines in ms):
Title - describes the study very well.
Abstract - well written but I suggest to add some numbers, e.g. concentration of spores in suspension, mortality of larvae, hatching rate in eggs.
Keywords - it is recommended not to repeat those which are in title, conidia dissemination might be replaced by more general "fungal spore dissemination"
Introduction - it would be good to add short overview of other biocontrol agents tested against the red palm weevil, e.g. entomopathogenic nematodes
40-41 and to the end of manuscript: citations should be in numerical format
61 genera name should be written in italics
75-76 full stop should be replaced with coma or semicolon and "To" replaced with "to" as all is part of a single sentence
92 could mesh size be specified with units?
104 not clear what termi "Sexed females" means, would it be enough to write just "Females" or "Mated females"?
105 please add name, city and country of manufacturer
106 typo in piece of sugarcane unit (instead of 1 cm2 should be volume, i.e. 1 cm3)?
110 were these pieces soaked on water before conidia powder was applied?
182 but according to text on line 112 counting started on day five?
217 two "eggs", should be corrected?
230-242 please be consistent with P symbol writing
Fig. 7 there seems to be some problem in B (letters B and probably part of C in middle of curves), I also suggest to delete statistics (e.g. line 280), they should be in Results only
309 also grooming
342 108 conidia mL-1 is not higher compared to 2–5 × 108 conidia mL-1
382-386 I have some doubts if this is true also for suspension treatment as we could expect that conidia get inside the foam when the foam is soaked in suspension, unlike the dry conidia powder which is mostly on surface, might require to comment this fact
393-394 could the difference between efficacy of formulations explained by different dose? Could dose for suspension be estimated (e.g. by volume of suspension absorbed by foam)?
Author Response
Reviewer 3:
Comments and Suggestions for Authors
The manuscript presents very nice study on interaction between one of the most serious pest of date and entomopathogenic fungi. The idea to apply conidia of fungus in the way that beetle female can transport them into place where it lays eggs is clever. Results are convincing and open new strategy for biocontrol of this pest. The manuscript for sure fits well into JoF scope, is well written, thought some parts can be improved. See comments below.
Specific comments (numbers indicate lines in ms):
Title - describes the study very well.
Abstract - well written but I suggest to add some numbers, e.g. concentration of spores in suspension, mortality of larvae, hatching rate in eggs.
Added accordingly.
Keywords - it is recommended not to repeat those which are in title, conidia dissemination might be replaced by more general "fungal spore dissemination"
Thank you. We removed the repetitive phrases and added other instead.
Introduction - it would be good to add short overview of other biocontrol agents tested against the red palm weevil, e.g. entomopathogenic nematodes
We added this to the introduction.
40-41 and to the end of manuscript: citations should be in numerical format
We have corrected accordingly.
61 genera name should be written in italics
Corrected.
75-76 full stop should be replaced with coma or semicolon and "To" replaced with "to" as all is part of a single sentence
As we added further explanation within these sentences, combining them into a single sentence will result in difficult to read phrase. We hope that with the modifications conducted, it is clearer now.
92 could mesh size be specified with units?
We have added mm to the mesh size (1.68 mm).
104 not clear what termi "Sexed females" means, would it be enough to write just "Females" or "Mated females"?
Thank you, we have changed the term throughout the entire text, from sexed to mated.
105 please add name, city and country of manufacturer
Added accordingly.
106 typo in piece of sugarcane unit (instead of 1 cm2 should be volume, i.e. 1 cm3)?
Corrected.
110 were these pieces soaked on water before conidia powder was applied?
Yes, we have added this to the text to make it clear.
182 but according to text on line 112 counting started on day five?
We corrected the sentence in the statistical analysis. We have counted the egg hatching every day, starting at day 5. However, no larvae were observed at day 5 (when we started the monitoring), thus we were able to state that there was no hatching from day 1 to 5.
217 two "eggs", should be corrected?
Yes, corrected.
230-242 please be consistent with P symbol writing
Corrected throughout the entire SM.
Fig. 7 there seems to be some problem in B (letters B and probably part of C in middle of curves), I also suggest to delete statistics (e.g. line 280), they should be in Results only
These two letters were removed, and the statistics were removed from the legend (but left in the main text).
309 also grooming
Added.
342 108 conidia mL-1 is not higher compared to 2–5 × 108 conidia mL-1
The magnitude was corrected (to 10^7, Gindin 2006).
382-386 I have some doubts if this is true also for suspension treatment as we could expect that conidia get inside the foam when the foam is soaked in suspension, unlike the dry conidia powder which is mostly on surface, might require to comment this fact
As was observed by confocal microscopy, before weevil entrance – the suspension treatment indeed resulted in the presence of conidia within the foam (Fig 2. G) but in very limited quantities. Moreover, calculating the GFP intensity demonstrated that there is no significant presence of conidia inside the foam treated with suspension as compared to dry conidia (Fig 6).
393-394 could the difference between efficacy of formulations explained by different dose? Could dose for suspension be estimated (e.g. by volume of suspension absorbed by foam)?
This is good point, but was not present in the text. We soaked the foam in 10 ml suspension of 1X10^8, and observed less than 5 ml suspension left after soaking (this was not counted in all repeats), resulting in around 5 to 8 ml of suspension used per each foam piece. We have added this data to the method section 2.3 (line 114).

Round 2
Reviewer 2 Report
Authors have made substantial changes and the clarity was much improved. However, the authors should explain why they used spores from SDA for spore suspension while using spores on rice for dried spores! This is important since spores from different media could have different virulence.
Author Response
Dear reviewer,
Following the important comment raised by you we have re-checked the methodological processes used in the study for conidial spores and powder preperations. We have realized that the description was confusing and not clearly describing the procedures.
Therefore we have revised fungal preparation accordingly in the revised manuscript.
The manuscript with the required amendments in track changes is attached.
Best regards,
Dana